

# Multi-variable evaluation of land surface processes in forced and coupled modes reveals new error sources to the simulated water cycle in the IPSL climate model

Hiroki Mizuochi[1, 2], Agnès Ducharne[2,3], Frédérique Cheruy[3,4,5], Josefine Ghattas[3], Amen Al-Yaari[2,3,6],
Jean-Pierre Wigneron[6], Philippe Peylin[3,7], Fabienne Maignan[3,7], Nicolas Vuichard[3,7]

[1]National Institute of Advanced Industrial Science and Technology (AIST), Geological Survey of Japan, Tsukuba, 305-8567, Japan
[2]UMR METIS (Milieux environnementaux, transferts et interactions dans les hydrosystèmes et les sols), Sorbonne Université, CNRS, EPHE, Paris, France
[3]IPSL (Institut Pierre Simon Laplace), Sorbonne Université, CNRS, Paris France
[4]LMD (Laboratoire de Météorologie Dynamique), Sorbonne Université, ENS, PSL Université, École polytechnique, Institut Polytechnique de Paris, CNRS, Paris France
[5]ISMAR/CNR Via del Fosso del Cavaliere, 100 00133 ROMA Italy
[6]INRAE, UMR 1391 ISPA, Villenave d'Ornon, France
[7]LSCE (Laboratoire des Sciences du Climat et de l'Environnement), UMR 8212 CEA-CNRS-UVSQ, 91191 Gif-sur-Yvette CEDEX, France

*Correspondence to:* Hiroki Mizuochi (mizuochi.hiroki@aist.go.jp)

**Abstract.** Evaluating land surface models (LSMs) using available observations is important to understand the potential and limitations of current Earth system models in simulating water- and carbon -related variables. To reveal the error sources of a land surface model (LSM), four essential climate variables have been evaluated in this paper (i.e., surface soil moisture, evapotranspiration, leaf area index, and surface albedo) via simulations with IPSL LSM ORCHIDEE (Organizing Carbon and Hydrology in Dynamic Ecosystems ), particularly focusing on the difference between (i) forced simulations with atmospheric forcing data (WATCH-Forcing-DATA-ERA-Interim: WFDEI) and (ii) coupled simulations with the IPSL atmospheric general circulation model. Results from statistical evaluation using satellite- and ground-based reference data show that ORCHIDEE is well equipped to represent spatiotemporal patterns of all variables in general. However, further analysis against various landscape/meteorological factors (e.g., plant functional type, slope, precipitation, and irrigation) suggests potential uncertainty relating to freezing/snowmelt, temperate plant phenology, irrigation, as well as contrasted responses between forced and coupled mode simulations. The biases in the simulated variables are amplified in coupled mode via surface–atmosphere interactions, indicating a strong link between irrigation–precipitation and a relatively complex link between precipitation–evapotranspiration that reflects the hydrometeorological regime of the region (energy-limited or water-limited) and snow-albedo feedback in mountainous and boreal regions. The different results between forced and





coupled modes imply the importance of model evaluation under both modes to isolate potential sources of uncertainty in the

model.

## 1 Introduction

Land surface models (LSMs) are essential to understand large-scale exchange of energy, water and carbon between the land surface and the atmosphere. LSMs coupled with atmospheric general circulation models (GCMs) have been used to simulate

global climate and climate change under international frameworks such as the Coupled Model Intercomparison Project (CMIP) (Taylor et al., 2012; Eyring et al., 2016a), contributing to Earth sciences and policy making for mitigating and adapting to climate change. To understand the potential and limitations of climate change simulations, evaluating outputs of LSMs with available observations is important (Flato et al., 2013). Uncertainties associated with LSMs can arise from a deficiency of model physics and parameterization (Liu et al., 2003), errors in atmospheric forcing data (Guo et al., 2006;

Nasonova et al., 2011; Yin et al., 2018), boundary conditions including vegetation and land use changes (Guimberteau et al., 2017; Boisier et al., 2014), and/or error propagation through land–atmosphere coupling (so-called "climate drift") (Dirmeyer, 2001). Recently, convenient tools for systematic model evaluation have been developed (e.g., Eyring et al., 2016c; Gleckler et al., 2016; Best et al., 2015); however, further in-depth model evaluation is required to reveal the underlying processes and sources that lead to uncertainties in simulations (Eyring et al. 2016b).

Notably, focusing on the differences between LSM simulations with and without GCM coupling would provide novel knowledge about LSM evaluation (Liu et al., 2003; Zabel et al., 2012; Wang, T. et al., 2015). LSM simulations without GCM coupling but forced by an atmospheric dataset (also called "offline" or "stand-alone" mode) do not allow feedback between the atmosphere and land surface. Therefore, errors in the simulated values solely arise from deficiency model structure/parameterization, uncertainty in the forcing data (Yin et al., 2018), and mismatch in land cover between

model and forcing data (Zabel et al., 2012). The foremost influential forcing factor on water cycle is precipitation (Qian et al., 2006; Decharme and Douville 2006), although radiation and land cover (i.e., vegetation) can also affect hydrological variables (Dirmeyer, 2001; Guo et al., 2006) such as surface soil moisture (SSM) or evapotranspiration (ET), depending on the temporal scale (Guo et al., 2006) and the hydrometeorological condition of the region (i.e., energy-limited or water-limited, Nasonova et al., 2011; Zabel et al., 2012). Anthropogenic factors (e.g., irrigation) may also cause errors in the

simulated variables when not accounted for by the LSM (Yin et al., 2018). On the other hand, coupled LSM simulation are also affected by errors in atmospheric simulation, which can be enhanced through land–atmosphere interaction (Mahfouf et al., 1995; Liu et al., 2003; Wang, T. et al., 2015). Such errors occur at short time scale (i.e., several-days) up to seasonal time scale (Dirmeyer, 2001), via the interlinkage of hydrological variables (e.g., rainfall, SSM, ET, and infiltration) in the LSM scheme and thermal variables (Cheruy et al., 2017, AitMesbah et al., 2015).





Among various LSMs, we focused on the Organizing Carbon and Hydrology in Dynamic Ecosystems (ORCHIDEE) LSM (e.g., Krinner et al., 2005; d'Orgeval et al., 2008; Guimberteau et al., 2017), which enables the explicit representation of processes governing the water, carbon, and energy budgets with highly flexible spatial resolution (Raoult et al., 2019). We used the ORCHIDEE (revision 4783, tag 2.0) version, which is implemented in the IPSL's (Institute Pierre Simon Laplace) climate model configurations used for CMIP6 (Eyling et al., 2016a), including the Land Surface, Snow, and

Soil Moisture Model Intercomparison Project (LS3MIP) with offline simulations (van den Hurk et al., 2016). Through an in-depth assessment of four simulated variables (i.e., SSM, ET, leaf area index (LAI), and surface albedo) that should be closely interlinked and a special focus on the differences between the forced and coupled simulations, the aim of this study is to better understand which land surface processes deserve further improvements in the studied LSM and to investigate the land–atmosphere coupling role in diagnosed model uncertainties.

The Global Climate Observing System (GCOS, 2010) designates the four selected variables as being essential climate variables (ECVs), thereby allowing us to take advantage of recent progress in their global-scale observation. Using satellite data, researchers have developed various retrieval algorithms to acquire SSM (Jackson et al., 1999; Wigneron et al., 2007, 2017), ET (Zhang et al., 2010; Miralles et al., 2011; Zeng et al., 2014), LAI (Zhu et al., 2013), and albedo (Schaaf et al., 2002; Qu et al., 2014), which can be used as reference data for LSM evaluation. Empirical upscaling products from

global *in situ* observations (Jung et al., 2011, 2019) can also be used. The selected variables are particularly interesting for land surface processes: SSM is a recognized driver of surface–atmosphere interactions (Seneviratne et al., 2010), constraining the partitioning of sensible/latent heat and plant activity and determining ET and vegetation dynamics (e.g., Gu et al., 2006). ET affects atmospheric humidity (usually described by the vapor pressure deficit) and cloud formation, creating feedback systems among SSM, ET, and precipitation (Yang et al., 2018). Accounting for long-term vegetation dynamics,

which can be measured by LAI, interlinked with such hydrological processes, is important in monitoring carbon cycle and ecosystem services that are related to climate change (IPCC, 2014) and natural disasters (Adikari and Noro 2010). Another important parameter in the surface energy exchange is the surface albedo, which controls the reflection of incident solar radiation and is interlinked with hydrological processes (especially through surface snow cover) and vegetation dynamics (Bonan, 2008).

To investigate the potential sources of model uncertainty, we considered various landscape factors ("factor analysis") in addition to the traditional statistical evaluation. This work aims at increasing knowledge about the features and limitations of ORCHIDEE and is a practical example of in-depth model evaluation focusing on the differences between forced and coupled modes. The remainder of this paper is organized as follows. Section 2 describes the simulation setting, the reference datasets, and the factor analysis. Section 3 presents results for the spatiotemporal patterns of the model

uncertainties and factor analysis. Finally, Sections 4 and 5 provide a discussion and conclusions, respectively.





## 2 Materials and methods

### 2.1 Model and simulations

#### 2.1.1 Description of the land surface model

ORCHIDEE (ORganizing Carbon and Hydrology in Dynamic EcosystEms) is the LSM used in the IPSL Earth System
model (ESM). This global process-based model of the land surface describes the complex links between the terrestrial
biosphere and the water and the energy and carbon exchanges between the land surface and the atmosphere (Krinner et al.,
2005). The used version in the IPSL-CM6 ESM for the CMIP6 simulations (Boucher et al., 2020), which is known as tag 2.0,
was previously described in many papers (Raoult et al., 2019; Boucher et al., 2020; Cheruy et al., 2020; Tafasca et al., 2020),
and we only summarize its main features in this paper, with some details on the related parametrizations to the four studied
ECVs.

        The land cover is described with 15 plant functional types (PFTs), including one for the bare soil, as seen in the full
list in Table 2, and they can all coexist in each grid-cell, where the taken fractions here are from the CMIP6 datasets
(Boucher et al., 2020). For each PFT, the transpiration serves as a coupling flux between the water, energy budget, and
photosynthesis process, which derive the evolution of the biomass and LAI owing to the generic equations with the PFT-
specific parameters (Krinner et al., 2005). Evapotranspiration (ET) is controlled by the energy and water budget via a bulk
aerodynamic approach, where four parallel fluxes are distinguished: sublimation, interception loss, soil evaporation, and
transpiration. In each grid-cell, the first two fluxes proceed at a potential rate from the grid-cell fractions with the snow and
canopy water, respectively. The soil evaporation and transpiration originate from the complementary snow-free fractions
covered by the bare soil and vegetation, which depend on the LAI, where the effectively covered fraction by the foliage
exponentially increases with the LAI with a coefficient of 0.5, while the light extinction is controlled through the canopy,
hence the photosynthesis process. The two fluxes both depend on the soil moisture, where the transpiration is limited by the
stomatal resistance, as it increased when the soil moisture dropped from the field capacity to the wilting point. The soil
evaporation is not limited by the resistance but only by the upward capillary fluxes, which control the soil propensity to meet
the evaporation demand.

The soil moisture (SM) dynamics are described over a soil depth of 2 m and discretized into 11 soil layers to solve
the Richards equation. The hydraulic conductivity and retention properties depend on the SM owing to the Van Genuchten-
Mulaem equations, with the parameters depending on the soil texture (Tafasca et al., 2020), and they are read from the map
of Zobler (1986). The infiltration is limited by the surface hydraulic conductivity, and it is calculated with a time splitting
procedure inspired by the Green-Ampt equation, where a sharp piston-like wetting front is assumed (d'Orgeval et al., 2008;
Vereecken et al., 2019). The surface runoff is made of non-infiltrated water (infiltration-excess runoff); however, ponding is
allowed in flat areas, and it can reinfiltrate at later time steps. This so-called reinfiltration fraction linearly decreases from 1
to 0 in totally flat grid-cells, where the mean grid-cell slope exceeds 0.5% (Ducharne et al., in prep). For the CMIP6, the





ORCHIDEE does not include the irrigation effect on the soil moisture, ET, and vegetation growth, although the model can simulate this anthropogenic pressure (Xi et al., 2018).

The snow processes are described using a 3-layer scheme of intermediate complexity (Wang et al., 2013), in which the snow albedo and insulating properties depend on the snow density and age. The ORCHIDEE 2.0 also includes a revised parameterization of the interplay between the vegetation and the snow albedo, and the optimized parameters match the remote sensing albedo data from the MODIS sensor, distinguishing the visible and near-infrared (NIR) bands (Boucher et al., 2020; Peylin et al., in prep). For the calculation of the heat diffusion, which includes the soil freezing effects (permafrost),

the soil is extended to 90 m, and the moisture content of the deepest hydrological layer is extrapolated to the entire profile between 2 and 90 m. The thermal soil properties depend on the soil texture, moisture, and carbon content (Guimberteau et al., 2018).

### 2.1.2 Forced and coupled simulations

To separate the errors caused by the ORCHIDEE model structure/parameterization from the ones resulting from the simulated climate through land–atmosphere coupling, we compared a forced and a coupled simulation. In the coupled simulation, the ORCHIDEE LSM is coupled to the LMDZ6A atmospheric GCM (Hourdin et al., 2020), as embedded in the IPSL-CM6 ESM for the CMIP6 simulations (Boucher et al., 2020). The coupled simulation was run over 1985–2014 (following a 5-yr spin up) using a 'nudging' approach to constrain the large-scale atmosphere dynamics toward the synoptic

atmospheric conditions (Cheruy et al., 2013). To this end, the simulated wind fields (zonal and meridional wind components) are relaxed toward the ERA-Interim winds (Dee et al., 2011) by adding a correction term in the evolution equation for the wind. By reducing the internal variability, this method allows the direct comparison of the observations and simulations, and it was successfully used for evaluating the coupled land–atmosphere parameterizations (Cheruy et al., 2013; Wang, et al., 2016), including the IPSL-CM6 ESM (Cheruy et al., 2020).

In the forced simulation, the required near-surface meteorological data by the ORCHIDEE LSM (liquid and solid precipitation, incoming longwave and shortwave radiation, 2-m air temperature and specific humidity, 10-m wind speed, surface pressure) are prescribed from the downscaled and bias-corrected reanalysis data [WATCH-Forcing-DATA-ERA-Interim (WFDEI)], provided at the 0.5° resolution with a 3-hourly time step (Weedon et al., 2011; Weedon et al., 2014). Precipitation is bias-corrected using monthly data from the Global Precipitation Climatology Centre (GPCC, Schneider et al.,

2014) and the simulation covers 1979–2009.

The spatial resolution differs between the two simulations, reflecting the grid of the atmospheric data: the coupled simulation has a coarser resolution (144 × 142, corresponding roughly to 2.5° in longitude and 1.25° in latitude) than that of the forced simulation (0.5° grid). To make the evaluation consistent and simple, we used the same spatial resolution for our analyses, and we oversampled the LMDz grid mesh to the finer resolution (0.5°) such as to keep as much spatial information

as possible from the high-resolution offline grid mesh. To investigate variability patterns on seasonal to interannual scales,




all the data were aggregated into monthly time steps. Four interlinked variables (SSM, ET, LAI, and albedo) were considered in this evaluation, and the study region was 60°S–90°N, 180°W–180°E (i.e., Antarctica and Greenland were excluded).

## 2.2 Reference data

### 2.2.1 Surface soil moisture

The SSM product provided by European Space Agency Climate Change Initiative (ESA CCI) (Liu et al., 2012) was used as a reference. It is a merged product comprising multiple SSM data derived from various passive and active microwave satellites (i.e., SMMR, SSM/I, TMI, AMSR-E, Windsat, SMOS, AMSR2, AMI-WS, ASCAT-A, and ASCAT-B) providing a long-term (1979–2015) SSM dataset with 0.25° resolution. The CCI-SSM product has been evaluated extensively against *in situ* observations (e.g., Al-Yaari et al., 2019b), and their accuracy has been reported as being relatively high compared to that

of other existing products such as SMOS-L3, LPRM, and AMSR2 (Ma et al., 2019).

Because it includes low-quality data flags for snow, dense vegetation, and radio-frequency interference (RFI) (Oliva et al., 2012), we applied data screening following Al-Yaari et al. (2016). We screened out all the pixels where the provided uncertainty was larger than 0.06 $m^3/m^3$ (volumetric water content). Next, any data records in which the SSM was not in a valid range (either >0.6 or <0.0) (Fernandez-Moran et al., 2017; Dorigo et al., 2013) were excluded. Finally, to exclude any

areas covered by snow or dense vegetation and other unreliable regions, we kept only those areas in which the quality flag was zero (fine-quality pixels). The screened dataset was then aggregated into 0.5° × 0.5° and monthly time steps. This screening process removed 3.6% of all the original pixels.

We performed an initial check on the time series of the global average of CCI-SSM and found an artificial trend therein that depended on the availability of the observation data (Supplementary Fig. S5). As reported by other researchers

(e.g., Loew et al., 2013), this artificial trend could lead to misinterpretation of long-term signals. To mitigate such artificial trends and initialization errors of each data, we selected a stable period (i.e., without discontinuous jump in the time series) during 1993–1999 for comparisons with both the forced and coupled simulations. Because of the differing natures of LSM-simulated and observed SSM (e.g., dependence on meteorological forcing data/atmospheric model, model parameterization), their absolute SSM [$m^3/m^3$] values (i.e., magnitudes) are not comparable (Reichle et al., 2004). In addition, since the CCI-

SSM product is scaled by the comparison with a different LSM (GLDAS-Noah), a direct comparison between the CCI-SSM and ORCHIDEE may lead to misleading results, as they have different soil representations (Raoult et al., 2019). Given this issue, the LSM- and satellite-based SSM were compared with statistically normalized values rather than absolute values of SSM (e.g., Polcher et al., 2016). Therefore, a spatiotemporal normalization (Equation 1) was applied to each co-masked dataset to eliminate systematic biases among the datasets and make the comparison reliable (Supplementary Fig. S5B):

$$SSM_{norm} = \frac{SSM - \overline{SSM}}{\sigma_{SSM}}, (1)$$

where $SSM_{norm}$ is the normalized SSM, and $\overline{SSM}$ and $\sigma_{SSM}$ are the mean and standard deviation, respectively, of all the available SSM sampled along spatial and temporal dimension during the period.





### 2.2.2 ET

In a preliminary study, we compared a ground-based machine learning ET product (Jung et al., 2011; 2019), three remote-sensing-based physical model products (Miralles et al., 2011; Zhang et al., 2010; Zeng et al., 2014), and their ensemble (see Supplementary Figs. S2, S7). We found that they showed similar spatiotemporal structures although they differed in absolute values in some regions, and the ground-based product was the most consistent with the ensemble. Therefore, we decided to use the ground-based ET (mm/d) as a representative, from 1987 to 2009. It is also advantageous in that it is derived from

upscaling of FLUXNET data (Jung et al., 2011; 2019) and is independent from specific ET-retrieval algorithms. The original spatial resolution (1°) of the data was resampled into 0.5° resolution to match that of forced simulation, and original temporal resolution was monthly time steps. A preliminary check of the time series and spatial patterns of the reference data revealed no artifact patterns (e.g., no abrupt jump in time series as found in CCI-SSM), so we used them with no pixel screening or normalization.

### 2.2.3 LAI


We used the global LAI dataset of Zhu et al. (2013), referred to hereinafter as LAI3g, which is based on a neural-network algorithm in conjunction with the third-generation Global Inventory Modeling and Mapping Studies (GIMMS 3g) and Moderate-resolution Imaging Spectroradiometer (MODIS) LAI product, with an original spatial resolution was 0.5° and half-monthly temporal resolution. Considering the common period among LAI3g and the coupled/forced simulations, we

selected 1987–2009 as the comparison period, and we resampled all data at 0.5° spatial resolution and aggregated them into monthly time steps.

### 2.2.4 Albedo

We used the MODIS albedo product (Qu et al., 2014) as reference data, which provided the bi-hemispherical reflectance (white-sky albedo) for the visible and NIR bands. The original 500-m spatial resolution and 16-d temporal resolution were

resampled (i.e., upscaled) into 0.5° resolution and monthly time steps. The common period between simulations and observation, 2003–2009, was used for evaluation. The pixels with retrieval failure of albedo were excluded from the analysis.

### 2.2.5 Precipitation

In addition to the four ECVs, we also evaluated the simulated precipitation because it is the primary factor that influences the hydrological variables (Qian et al., 2006; Decharme & Douville, 2006). In addition, we used the GPCC dataset, which was

also used to construct the meteorological forcing of the offline ORCHIDEE simulation. This gridded product at 0.5° provides monthly precipitation derived from quality-controlled observed precipitation from world-wide stations (Schneider et al., 2014). As this is the forcing data, the precipitation output in the forced simulation is identical to the reference. Therefore, the model evaluation regarding the precipitation was only conducted for the coupled simulation.





**2.2.6 Data processing**

For consistency between the observed and simulated data, we subjected the former to aggregation or resampling toward the 0.5° spatial resolution and monthly time steps for each variable, as described above. Due to the presence of data gaps in the reference data sets, which are either because of the acquisition issues or the quality control and data screening, we masked the simulated datasets to match the spatial-temporal data availability of the corresponding reference data. For the SSM, the

dense snow regions (with a snow water equivalent exceeding 48 mm) in the simulated data were further excluded so as to avoid unreliable comparisons with uncertain references. Also, co-masking was performed after the spatiotemporal resampling, followed by statistical normalization (only for the SSM). The resulting coverage of the selected comparison period is summarized in Table 1 for each variable.

After the above-mentioned pre-processing, to compare the spatial patterns of the observed and simulated data, we

focused on three accuracy criteria calculated at the 0.5° scale along the monthly time steps: the bias, correlation coefficient (CC), and root-mean-square error (RMSE). The criteria were calculated along the temporal axis for each pixel (i.e., the result was shown as one global map for a criterion). Note that the evaluation periods were different among the SSM (1993–1999), ET, LAI, precipitation (1987–2009), and Albedo (2003–2009). However, the impact of the chosen period on the evaluation is likely to be limited (see Supplementary Table S1).

**2.3 Factor analysis**

To reveal features of the simulations in detail, the accuracy criteria were evaluated against various landscape/meteorological factors (Figure 1), namely PFT, LAI, irrigation, precipitation, slope, snow, and ET. For each factor, time series were averaged temporally to make only one global map (i.e., the classification criteria were applied on long-term basis). The value of each factor was classified into a specific number of levels (classes), which were used as ordinal scales. Each factor was

classified as given in Table 2, and each factor is described in detail below:

1) For PFT, we used the input dataset that is used in ORCHIDEE. This includes fractional coverages in each pixel of 15 PFTs. We created a dominant PFT map by picking up the PFT class that have maximum fractional coverage for each pixel.

2) For LAI, we used the LAI3g data (above subsection), classifying them into three levels (see Table 1 for the specific class definitions).

3) For irrigation, we used a global map of irrigation areas (Siebert et al., 2010), which indicates the fractional coverage (%) of an irrigated area with 5-arc-min spatial resolution. It was classified into six levels.

4) For precipitation, we used the pluri-annual mean of GPCC during the same period as the investigated ECVs. It was classified into five levels.

5) For slope, this classification was done by referring to the ETOPO DEM (1 arc-minute global relief model of Earth's

surface; Amante & Eakins, 2009), which is also used in ORCHIDEE to control reinfiltration of the water.





6) For SWE, we used the pluri-annual mean of the corresponding ORCHIDEE SWE, which was classified into five levels.

7) For ET, we used the pluri-annual mean of Jung et al. (2011; 2019) during the same period as the investigated ECVs.

## 3 Results

### 3.1 Spatial and temporal patterns of model errors

Overall, the spatial structures of the ECVs simulated in both modes were consistent with those of the references (Supplementary Figures S1-S4; Fig. 1A). Figure 2 shows the spatial bias patterns of the four variables (normalized SSM, ET, LAI, and albedo) in both forced and coupled mode, and of the precipitation in coupled mode. Spatiotemporal averages of bias, RMSE, and correlation coefficients are summarized in Table 3. The spatial patterns of the temporal CC are also shown for SSM and albedo (Figure 3) for further discussion.

Figure 2A–C showed that the spatial pattern of normalized SSM bias in forced and coupled modes were consistent and delineated the biased regions clearly. The strong negative biases in normalized SSM was observed over the boreal region (except Eastern Siberia) with high SWE values (Figure 1D), suggesting the relation to snow or permafrost. Note that satellite observation uncertainties in such snowy regions could also be a reason for the discrepancy. The farm belt of India and China (with a lot of irrigation in Figure 1C) exhibit a systematic lower bias in SSM. Apart from those, arid (North Africa, middle of

Australia, north China) and tropical (Congo and Amazon Basin) regions also showed lower correlation (Figure 3A–C), part of which can be attributed to the inherent feature that CC tends to be low when the range in which the sample varies is narrow. To better identify the error sources in SSM, we plotted the mean seasonal cycles (i.e., monthly climatology) separately for each latitude zone (Figure 4). Substantial parts of the time series were consistent between simulation and observation (except grayed-out period due to insufficient sample size and low reliability of the reference data). The

underestimated simulated SSM values compared to the CCI-SSM values in the summer season in 30–60°N (Figure 4B) may be attributed to anthropogenic water input due to irrigation because this region includes large-scale agricultural fields (Figure 1C). In the low-latitude regions, the simulated values tend to underestimate SSM in the dry season, and to show larger seasonal change (Figure 4C, D).

     Most of the areas exhibited small ET biases in absolute value (Figure 2E, F), suggesting that ORCHIDEE is highly

capable of representing global ET. The coupled simulation tended to simulate larger ET values than did the forced simulation, which can be explained to some degree by the precipitation biases in the coupled simulation, which are positive on average (Table 3). Regions with large ET biases were distributed in the tropical (Amazon and Congo Basin, the maritime continent), mountainous (the Rockies, Andes, and Himalayas), and agricultural (especially in India) regions. Mountainous regions tended to be characterized by a positively biased precipitation simulation (Figure 2C), which caused positive bias of

ET in the coupled simulation (especially in North/South America). Tropical regions exhibited complex responses in ET between the coupled and forced simulations. The maritime continent (Indonesia and the other tropical Pacific islands) had negative ET biases for both simulations. Congo and a large part of the Amazon exhibited contrasting patterns between the





simulations (the uncoupled one had a negative bias whereas the coupled one had a positive bias). The link between the coupled ET simulation and the simulated precipitation was only straightforward in the Congo, i.e., the positively biased precipitation (water input) led to the positive bias of ET. In the maritime continent, the coupled ET was negatively biased despite the positive bias of precipitation. By contrast, the coupled ET was positively biased in the Amazon despite the negative bias of precipitation.

Positive bias of LAI was observed in large areas globally (Figure 2F, G). Given the strong similarity between the forced and the coupled bias maps, it is suggested that the bias comes mostly from the surface component, such as PFT maps, or reference data itself. In fact, LAI retrievals by spaceborne sensors like MODIS may be saturated for large values of LAI (Zhao et al., 2016), resulting in underestimation of LAI in reference. Despite such a positive-bias tendency, the boreal region in Eastern Siberia, the shores of the Great Lakes in North America, and the basin of the Mekong River all exhibited negative bias of LAI. In addition, there were hotspots of negatively biased LAI in such regions as the Zambezi River system lying across Angola and Zambia. Contrasting biases between the simulations were observed around the Himalayas.

In most regions, a consistent spatial pattern with reference was obtained for total albedo, with showing strong spatial similarity between the forced and the coupled modes (Figure 2H, I). The largest biases were the overestimation in the mountainous regions (especially the Himalayas in the coupled mode) and the underestimation in the boreal and polar regions, where snow affects the albedo. In addition, simulated and observed albedo were uncorrelated (or negatively correlated) in many regions apart from the boreal one (Figure 3C–F). Low correlation coefficients in the arid and tropical region can be attributed to the temporal invariance of the land surface. However, even in some temperate and semi-arid zones where temporal variance is likely to be high, low correlation was observed. In such regions, seasonal changes of the land surface (caused mainly by vegetation phenology and the snowfall/snowmelt cycle) may not be described well in ORCHIDEE. In fact, the global monthly climatology (Figure 5A, F) showed a global mean overestimated NIR albedo in JJA and underestimated visible albedo in MAM. The main source of the NIR albedo overestimation seemed to be that in the temperate zone (30–60°N; Figure 5C), suggesting an overestimated vegetation cover (having high reflectance in the NIR spectral region) there in the summer. Also, there was a systematic overestimation of the albedo in the tropical band (Figure 5D, E, I, J) and a small underestimation in the snow-related season (winter to spring) of the boreal band (Figure 5B, G).

### 3.2 Factor analysis

The bivariate linear regressions between the simulated ECV bias and the factors (Table 4) and the boxplots against each factor class (Figures 6-8) first revealed a large bias variability within each class, resulting in a large part from the spatial variability of the simulated variables across the various climates and biomes of the globe. However, some controls could be identified despite this variability. It is particularly the case for irrigation, which has an obvious impact on the simulated hydrological variables (SSM, ET, LAI, and precipitation; Figure 6A, C, E, G): both the coupled and forced models show negatively biased values in the largely irrigated areas (classes 5 and 6), except for the forced-mode SSM. This is understandable because the simulations overlook irrigation, which creates artificial water input to the soil, resulting in





additional ET and plant growth in reality. Interestingly, the coupled simulation underestimates the observed values more than does the forced one (Figure 6A, C, E, G), which probably relates to a positive feedback driven by surface–atmosphere coupling (Mahfouf et al., 1995; Liu et al., 2003; Wang et al., 2015). Since the forcing precipitation (i.e., real-world precipitation) integrates the impact of real-world irrigation, this factor has a relatively weak effect in the forced mode.

The contrasting ET-bias pattern between forced and coupled modes in the Congo and the Amazon (Figure 2D, E) was also confirmed in the factor analysis of precipitation (classes 4 and 5, which probably correspond to tropical regions; Figure 6D), PFT (class 2: broadleaf evergreen in Figure 7A), LAI (class 3 in Figure 7B), and ET (class 3 in Figure 8A). This also explains the contrasting correlation sign of ET bias with P, SSM, ET and LAI in Table 4.

    The factor analysis confirms the positive bias of LAI in the tropical regions, which are characterized by high

precipitation (classes 4 and 5 in Figure 6F), broadleaf evergreen forest (PFT 2 in Figure 7C), high LAI (class 3 in Figure 7D), and high ET (class 4 in Figure 8B). However, some of the positive bias in such tropical regions might be compensated by the negative bias of the simulated precipitation (especially in the Amazon; Figure 2C, also confirmed by class 3 in Figure 7J), resulting in a smaller bias of LAI in the coupled simulation than that in the forced simulation. Negative LAI bias in the boreal region is also confirmed by the PFT factor analysis (classes 8, 9, and 15 in Figure 7C).

For albedo, the effect appeared in the factor analysis against slope (class 3 in Figure 8C, D) and SWE (classes 4 and 5 in Figure 8F, G) as a discrepancy between the coupled and forced simulations. In the steep region, the coupled simulation tended to be positively biased because of the precipitation bias. In the high-SWE region, negative bias of albedo was enhanced in the forced simulation. This is due to the already mentioned compensation of the positive bias in the mountainous region with the negative bias in the boreal and polar regions (Figure 2H, I). The NIR albedo in the tropical region (classes 2

and 3 in Figure 7E and class 3 in Figure 7F) tended to be slightly large for both simulations. This is consistent with the positive bias of LAI in such regions (Figure 2F, G), although the range of bias was small.

## 4 Discussion

In general, the ORCHIDEE simulations show good spatial/temporal consistency with the reference data, except for issues related to external water addition/subtraction and surface–atmosphere coupling. An example of the external source of water

input is irrigation. Largely irrigated areas obviously lead to underestimated hydrology-related model parameters (i.e., SM, ET, and LAI). Although the impact of irrigation on ORCHIDEE SSM simulation has been suggested by Yin et al. (2019) over a specific region (China), our experiment demonstrated explicitly that the effect on SSM in the forced mode is relatively small on the global scale, and rather larger on ET and LAI (Figure 6A, C, E). Integrating the irrigation process in ORCHIDEE with an ancillary agricultural map and data assimilation (Raoult et al., 2019) may improve the accuracy (de

Rosnay et al., 2003). Through the land–atmosphere coupling (Al-Yaari et al., 2019a), the impact of the irrigation is emphasized in the coupled simulation (Figure 6A, C, E, G), where strong negative bias was observed in not only ET, LAI, and precipitation but also SSM over largely irrigated areas. Specifically, a lack of description of the additional water input





and man-made vegetation over irrigated agricultural land led to lower SSM and LAI, which in turn led to lower ET. In the coupled simulation, the lower SSM also led to lower humidity and lower precipitation, resulting in enhanced underestimation

of SSM in the next time step (i.e., positive feedback). The enhanced SSM underestimation caused enhanced ET underestimation, as well as enhanced LAI underestimation through the parametrizations of carbon assimilation and vegetation phenology. Underestimation of precipitation in the coupled simulation over irrigated areas (e.g., India in Figure 2C; classes 5 and 6 in Figure 6G) supports the validity of this scheme, and such an emphasizing effect in the coupled model is consistent with other reports (Mahfouf et al., 1995; Liu et al., 2003; Wang et al., 2015). The spatial similarity between the

bias maps of SSM, ET, and precipitation (Figure 2A–E) over central-south Africa, Australia, and a large part of south and east Asia also suggests the strong interlink between them in the coupled mode. This is consistent with the results of Yang et al. (2018), who have reported the positive feedback of SSM–precipitation and the positive correlation of SSM–ET and ET–precipitation in such transient zones (i.e., a climate that is neither extremely dry nor extremely wet).

However, there seem to be other secondary factors that should be considered regarding the hydrometeorological

regime (i.e., energy-limited or water-limited). In particular, ET is not controlled solely by precipitation but also by radiation (Cheruy et al., 2020), and temperature determines the potential ET (Dirmeyer, 2001; Nasonova et al., 2011). The complex response of ET to precipitation presented in the present study suggests the importance of those factors: in the coupled simulation, positive precipitation bias in the Congo Basin (Figure 2C) created positive ET bias (Figure 2E) in a straightforward manner. By contrast, there may be a negative feedback in the Amazon and the maritime continent between

precipitation and ET because these areas are strongly energy-limited (Seneviratne et al., 2010; McVicar et al., 2012) in comparison to the Congo. In the maritime continent, positive precipitation bias meant more cloud coverage than reality, which decreased the available energy and ET. Oppositely, negative precipitation bias in the Amazon meant less cloud coverage, larger available energy, and larger ET than reality.

Although such feedback explains the overestimated ET in the Congo and the Amazon in the coupled simulation, it

does not explain the underestimated ET there in the forced simulation. The potential reason for it is excessive water stress on ET in regions of high precipitation in the forced mode, although that is not clear in the coupled mode because of the positive P bias, which could cancel the negative ET bias. In addition, conversely, the too weak water stress in the dry areas (either for transpiration or soil evaporation) can also explain the negative correlation between the forced-mode ET bias and the precipitation (Table 4). A solution would be to activate a resistance to soil evaporation, increasing with the top soil dryness

(Cheruy et al., 2020). Such contrasting results between the forced and coupled modes imply the importance of model evaluation under both modes to isolate the potential error sources.

Compared with the forced mode, the positively biased precipitation simulated by the coupled mode may positively bias the albedo, particularly in the mountainous (Figure 8C–E) via considerable snow cover. This probably arose from incomplete atmospheric simulation of the local climate (Cheruy et al., 2020) such as an updrift along a mountain slope.

Coarse spatial resolution of the atmospheric simulation in the coupled mode can also make it difficult to represent the impact of mountainous topography on local climate (Decharme and Douville, 2006). Theoretically, the overestimation of albedo





should decrease the available energy at the surface, thereby decreasing ET and surface temperature. The slight negative bias of ET in the Himalayas (Figure 2E) despite the positive bias of precipitation (Figure 2C) can be explained by the decrease in available energy due to the increased albedo (Figure 2I). Such an ice–albedo interaction in the ORCHIDEE-LMDZ coupled

mode has also been reported over the boreal region (Wang, T. et al., 2015; particularly pronounced in spring temperature over Eastern Siberia). Taking the ice–albedo feedback into consideration with the secondary factors (i.e., radiation and temperature) that affect ET, the link between precipitation and ET in the coupled mode is rather complex in the mountainous and boreal regions. Moreover, the deficit of available energy may reduce photosynthesis thus vegetation growth, causing a peaky underestimation of LAI in the Himalayas in the coupled mode (Figure 2G), which is not observed in the forced mode

(Figure 2F).

Part of the positive biases in normalized SSM in the Eastern Siberia and polar region (Figure 2A, B) may be attributed to freezing/snowmelt and related vegetation phenology. Snowmelt that is considerable and/or very fast occurs in the spring in ORCHIDEE (Figure 5B, G), leading to overestimated SSM. However, there is likely to another control factor, such as wetlands, permafrost, and albedo. Underestimation of albedo in many boreal zones (Cheruy et al., 2020) was

expected to lead to overestimated ET, but it did not lead to an obvious ET bias because of the underestimated LAI (Figure 2F, G). Given the spectral features of land cover (Petty, 2006), the NIR albedo is related largely to an abundance of vegetation, i.e., LAI. Therefore, uncertainty in snow and LAI leads to uncertainty in the surface albedo, which further propagates uncertainty in the energy balance and water cycles. Such a complicated relationship should be treated in the special tuning of ORCHIDEE for high latitudes (Druel et al., 2017; Guimberteau et al., 2018). In addition to high-latitude regions, vegetation

seasonality in the temperate zones seemed uncertain. In the temperate forests, the model is likely to simulate spring green-up that is considerable and/or very fast (Figure 5C), which causes an overestimated NIR albedo and discrepancies in LAI and albedo seasonality.

Regardless of the origin (i.e., satellite, reanalysis, or *in situ*), observations inevitably contain inherent uncertainty, which leads to uncertainty in the model assessment. SSM retrieval over substantially high/low vegetation, tropical/arid

regions, and highly heterogeneous and high-roughness regions remains challenging (Ma et al., 2019). Therefore, some part of the low SSM correlation in arid/tropical regions (Figure 3A, B) can be attributed to uncertainties in the satellite products in addition to an inherent feature of CC. Snow cover and RFI (Oliva et al., 2012) may also cause uncertainties in satellite-based SSM estimation, although we attempted to remove such uncertain pixels by means of a preliminary quality check. Using multiple data sources (e.g., the Soil Moisture Active Passive (SMAP) product [Ma et al., 2019; Al-Yaari et al.,

2019b]) as reference for model evaluation (Eyring et al., 2016b) is a promising way to address such uncertainties. A brief attempt with SMOS-IC product (Fernandez-Moran et al., 2017) was shown in Supplementary Fig. S6. Inconsistency between the model-simulated SSM depth (up to 10 cm) and the penetration depth of satellite sensors (several centimeters) may also cause uncertainties in the assessment, although using normalized SSM instead of absolute SSM is likely to mitigate the effect to some extent.





The satellite-based LAI product (Zhu et al., 2013) may be affected by the saturation issue of optical satellite data (i.e., MODIS) in regions with high LAI. The snow albedo of the MODIS product (MCD43) has a slightly larger uncertainty (RMSE ≈ 0.07) (Stroeve et al., 2005; Stroeve et al., 2013) than that of the snow-free daily mean albedo (RMSE = 0.034) (Wang, D. et al., 2015). However, this does not alter our conclusion about the ORCHIDEE albedo uncertainty in the snow region, but some of the uncertainty might be attributed to the error in satellite observation.

We depended largely on satellite-derived data for the SSM, LAI, and albedo evaluations. By contrast, we used a FLUXNET-based product (Jung et al., 2019) for the ET evaluation, which has potential uncertainties arising from (i) the statistical upscaling process (model tree ensemble: Jung et al., 2009), (ii) the input data required in machine-learning prediction, and (iii) the heterogeneous distribution of ground stations. Because the latter potential issue is particularly important for hardly accessible regions such as tropical and mountainous areas, progress in the data coverage of the
FLUXNET network is desirable. Although ET products derived from satellite data (Miralles et al., 2011; Zhang et al., 2010; Zeng et al., 2014) can also be used, unlike the other variables (SSM, LAI, and albedo), the retrieval of ET is not done directly from the satellite observations but depends largely on the process-oriented models. Therefore, in addition to the uncertainties in the satellite observations themselves, such products have uncertainties that arise from ancillary data (e.g., atmospheric conditions, land cover) required in the model, as well as from imperfections in the model
structure/parameterization (preliminary comparison among the different data sources can be found in Supplementary Fig. S7).

     Note that the present study is based on a specific LSM (i.e., ORCHIDEE 2.0), atmospheric model (i.e., LMDZ6A), and forcing data (WFDEI). Future work should include addressing the uncertainties that arise from the LMDZ model structure/parameterization, as well as the resolution in the numerical simulation (Hourdin et al., 2013). Uncertainties that
arise from the atmospheric model have been analyzed for some evaporation and SSM by Cheruy et al. (2020). For China, WEDFI-based simulations have performed better than Princeton Global meteorological Forcing and Climatic Research Unit-National Center for Environmental Prediction with ORCHIDEE (Yin et al., 2018). However, because varying the forcing data has a comparable impact to varying the LSM in the forced simulation (Guo et al., 2006), the uncertainty in selecting the forcing data should also be kept in mind. Other future work should be factor analysis against other hydrometeorological
parameters such as radiation, temperature, and precipitation frequency (Qian et al., 2006; Yin et al., 2018).

## 5 Conclusions

This paper has presented an in-depth evaluation of four interlinked essential climate variables (namely surface soil moisture, evapotranspiration, leaf area index, and albedo) simulated by ORCHIDEE land surface model under different simulation modes (either forcing by WFDEI or coupled with LMDZ). Statistical evaluation was conducted using various reference-data
sources (ESA CCI, upscaled FLUXNET, GIMSS 3g and MODIS products), and factor analysis was conducted against various landscape factors (namely plant functional type, leaf area index, irrigation, precipitation, slope, snow water





equivalent, and evapotranspiration). Although ORCHIDEE consistently represented the spatiotemporal patterns of each essential climate variable in general, some issues were found relating to water cycles and their different consequences between the forced and coupled simulations. Errors relating to freezing/snowmelt, artificial water input such as irrigation,

and precipitation bias propagated through surface–atmosphere coupling in the coupled mode. The factor analysis revealed a strong link between irrigation and precipitation (that further affected surface soil moisture, evapotranspiration, and leaf area index, particularly in the coupled mode) and a relatively complex link between precipitation and evapotranspiration that reflected the hydrometeorological regime of the region (energy-limited or water-limited) and the snow-albedo feedback in mountainous and boreal regions. In addition, the description of vegetation and snow seasonality seemed to be an issue in

ORCHIDEE. Considerable and/or very fast green-up in temperate forest may lead an overestimation of leaf area index and near infrared albedo. Considerable and/or very fast snowmelt in spring in the boreal region may result in the underestimation of albedo in such regions, which can affect energy balance and water cycles. The different results between the forced and coupled modes stress the importance of model evaluation under both modes to determine each potential error source in model simulation.

**Acknowledgments**

This research was supported by the Japan Society for the Promotion of Science (JSPS KAKENHI; Grant number 16J00783) and by the Centre national d'études spatiales (CNES, under the program Terre Océan Surfaces Continentales et Atmosphère). It benefited from the ESPRI (Ensemble de Services Pour la Recherche l'IPSL) computing and data center (https://mesocentre.ipsl.fr), and it was supported by CNRS, Sorbonne Université, Ecole Polytechnique and CNES and by

other national and international grants. The SMOS and CCI products were obtained from the "Centre Aval de Traitement des Données SMOS" (CATDS) and the European Space Agency (ESA), respectively. The CMIP6 project at the IPSL used the HPC resources of the TGCC (Très Grand Centre de calcul du CEA) under the allocations 2016-A0030107732, 2017-R0040110492, and 2018-R0040110492 (project gencmip6), which were provided by the GENCI (Grand équipement national de calcul intensif).

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





**Table 1. Overview of the selected reference datasets and analysis period (The last column provides the percent of the land pixels in the global maps with the observed values. A smaller amount of the SSM data was available in comparison with the others as a result of the relatively strict quality control. For the ET and LAI, no data were available in the extremely arid regions.)**

| Variable | Reference product | Evaluation period | Observed fraction of land area (%) |
|---|---|---|---|
| SSM | ESA CCI v4.4 (Liu et al., 2012) | 1993-1999 | 44.1 |
| ET | Jung et al. (2019) | 1987-2009 | 89.3 |
| LAI | LAI3g (Zhu et al., 2013) | 1987-2009 | 87.8 |
| Precipitation | GPCC (Schneider et al., 2014) | 1987-2009 | 98.5 |
| Albedo | MODIS (Qu et al., 2014) | 2003-2009 | 87.4 |

**Table 2. Correspondence between classification levels and values for each factor.**

| Factor | Reference data | Classification | Fraction of the land area (%) |
|---|---|---|---|
| PFT | ORCHIDEE-defined plant functional types | class 1: Bare soil is dominant. | 15.4 |
| | | class 2: Tropical broadleaf evergreen forest is dominant. | 6.9 |
| | | class 3: Tropical broadleaf raingreen forest is dominant. | 3.2 |
| | | class 4: Temperate needleleaf evergreen forest is dominant. | 2.0 |
| | | class 5: Temperate broadleaf evergreen forest is dominant. | 3.4 |
| | | class 6: Temperate broadleaf summergreen forest is dominant. | 3.6 |
| | | class 7: Boreal needleleaf evergreen forest is dominant. | 8.6 |
| | | class 8: Boreal broadleaf summergreen forest is dominant. | 6.5 |
| | | class 9: Boreal needleleaf summergreen forest is dominant. | 8.6 |
| | | class 10: Temperate C3 grasses are dominant. | 6.6 |
| | | class 11: C4 grasses are dominant. | 8.8 |
| | | class 12: C3 crops are dominant. | 9.2 |
| | | class 13: C4 crops are dominant. | 2.1 |
| | | class 14: Tropical C3 grasses are dominant. | 2.7 |
| | | class 15: Boreal C3 grasses are dominant. | |



| | | | 12.4 |
|---|---|---|---|
| LAI | Zhu et al. (2013) | class 1 (low LAI): 0 – 1.0 m²/m² | 41.9 |
| | | class 2 (middle LAI): 1.0 – 3.0 m²/m² | 37.1 |
| | | class 3 (high LAI): 3.0 – m²/m² | 8.7 |
| ET | Jung et al. (2019) | class 1: less than 1 mm/d | 45.8 |
| | | class 2: 1–2 mm/d | 23.6 |
| | | class 3: 2–3 mm/d | 13.0 |
| | | class 4: more than 3 mm/d | 6.9 |
| Precipitation | GPCC, Schneider et al. (2014) | class 1 (extremely dry): less than 1 mm/d | 41.0 |
| | | class 2 (dry): 1 to 2 mm/d | 24.1 |
| | | class 3 (moderate): 2 to 4 mm/d | 17.2 |
| | | class 4 (wet): 4 to 7 mm/d | 11.0 |
| | | class 5 (extremely wet): more than 7 mm/d | 5.3 |
| SWE | ORCHIDEE-simulated SWE | class 1: 0 mm | 33.3 |
| | | class 2: 0–16 mm | 33.6 |
| | | class 3: 16–32 mm | 8.7 |
| | | class 4: 32–48 mm | 9.5 |
| | | class 5: more than 48 mm | 14.8 |
| Irrigated area | Siebert et al. (2010) | class 1: 0% | 56.6 |
| | | class 2: 0–5% | 34.7 |
| | | class 3: 5–10% | 3.6 |
| | | class 4: 10–20% | 2.6 |
| | | class 5: 20–50% | 1.9 |
| | | class 6: 50–100% | 0.5 |
| Slope | ETOPO (Amante & Eakins, 2009) | class 1 (flat): 0-0.5- degree | 3.0 |
| | | class 2 (middle): 0.5–2.0 degree | 28.1 |
| | | class 3 (steep): 2.0- degree | 67.4 |





**Table 3. Land averages of the evaluation criteria (bias, RMSE, and correlation coefficient CC) for the selected variables and reference data sets (The same bias sign on the average over land for all the variables were observed between the forced and coupled simulations. The positive systematic bias was observed for the LAI, and large uncertainty (i.e., RMSE) was observed for the SSM and ET. The ET shows the best correlation coefficient. Overall, the coupled simulation tends to behave more realistically, despite the overestimation of the precipitation.)**

| | | Forced | Coupled |
|---|---|---|---|
| Bias | Precipitation (mm/d) | - | 0.186 |
| | SSM (normalized) | -0.072 | -0.062 |
| | ET (mm/d) | -0.231 | -0.133 |
| | LAI (-) | 0.325 | 0.220 |
| | Albedo (-) | -0.000 | 0.009 |
| RMSE | Precipitation (mm/d) | - | 1.680 |
| | SSM (normalized) | 0.546 | 0.560 |
| | ET (mm/d) | 0.513 | 0.540 |
| | LAI (-) | 0.586 | 0.554 |
| | Albedo (-) | 0.048 | 0.047 |
| CC | Precipitation (mm/d) | - | 0.605 |
| | SSM (normalized) | 0.581 | 0.551 |
| | ET (mm/d) | 0.744 | 0.692 |
| | LAI (-) | 0.328 | 0.340 |
| | Albedo (-) | 0.395 | 0.426 |


**Table 4. Spatial correlation coefficients (SCC) between the biases (in the forced and coupled modes) and the potential explanatory factors (The PFT was excluded in the Table because it is a nominal scale. The statistically insignificant SCCs appear in italic. The SSM tended to be underestimated in the high P, SSM, ET, and LAI regions for both the forced and coupled modes. Between the forced and coupled ET, the opposite association with the P, ET, and LAI was observed. The LAI in both modes were positively biased in the high P, ET, and SSM regions (probably corresponding to the tropical region). The albedo and coupled P were strongly associated with the slope. The irrigation was likely to negatively bias the SSM and ET, and the effect was more enhanced in the coupled mode.)**

| | Biases of the forced simulations | | | | |
|---|---|---|---|---|---|
| Factors | P | SSM-CCI | ET | LAI | Albedo |
| P | - | -0.203 | -0.168 | 0.375 | 0.283 |
| ET | - | -0.163 | -0.277 | 0.357 | 0.344 |





| LAI | - | -0.083 | -0.127 | 0.263 | 0.275 |
|---|---|---|---|---|---|
| SWE | - | 0.024 | -0.103 | -0.090 | 0.181 |
| Irrigated fraction | - | -0.066 | -0.170 | 0.012 | 0.059 |
| Slope | - | -0.068 | -0.010 | 0.027 | 0.023 |
| | Biases of the coupled simulations | | | | |
| Factors | P | SSM-CCI | ET | LAI | Albedo |
| P | -0.108 | -0.234 | 0.200 | 0.258 | 0.130 |
| ET | *0.006* | -0.164 | 0.163 | 0.245 | 0.139 |
| LAI | *-0.004* | -0.121 | 0.264 | 0.092 | 0.103 |
| SWE | 0.034 | *-0.009* | -0.134 | -0.060 | 0.073 |
| Irrigated fraction | -0.071 | -0.118 | -0.213 | -0.012 | 0.030 |
| Slope | 0.267 | 0.085 | -0.022 | -0.031 | 0.249 |






**Figure 1: Spatial patterns of temporally averaged reference data used for factorial analysis: (A) GIMMS LAI; (B) Slope derived from ETOPO; (C) fractional area equipped with irrigation; (D) snow water equivalent derived from the forced-mode ORCHIDEE; (E) GPCC precipitation data (F) ET product provided by Jung et al; and (G) plant functional type used in ORCHIDEE.**










**Figure 2: Temporally averaged spatial patterns of bias (i.e., simulated values minus observed values) for the four variables (SSM, ET, LAI and albedo) and for the coupled precipitation. (A), (B): SSM bias between simulation and CCI-SSM in normalized SSM during period 1 (1993–1999) for forced and coupled mode, respectively. (C): that between simulation and SMOS-IC during period 2 (2011–2014) for coupled mode. (C): precipitation bias between coupled simulation and GPCC. (D), (F), (H): ET, LAI, total-albedo biases between simulated and observed values for forced mode, respectively. (E), (G), (I): those for coupled mode. Reference observations correspond to ET: upscaled FLUXNET data, LAI: LAI3g data, and albedo: MODIS albedo product. Gray areas are statistically insignificant pixels.**





Figure 3: Spatial patterns of correlation coefficient along time series (with monthly time steps) per pixel, for SSM and albedo. (A), (B): correlation coefficient between simulated SSM and CCI-SSM in normalized SSM for forced and coupled mode, respectively. (C), (D): correlation coefficient between simulated and observed (MODIS) albedo in NIR band for forced and coupled mode, respectively. (E), (F): those in visible band. Gray areas are null pixels that were excluded by the quality control, and green areas are statistically not significant pixels.



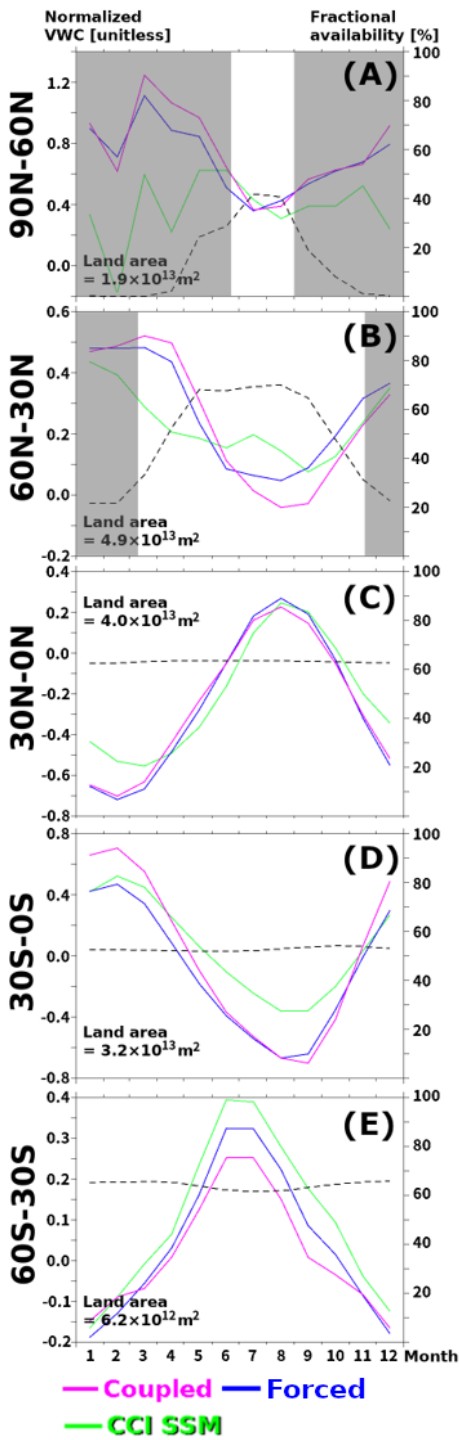

**Figure 4: Comparison of seasonal patterns among reference (CCI-SSM) and simulations (forced and coupled) for each latitude zone. Dashed black line is the fraction of available pixels to all land pixels over each zone. Depending on the snow mask, the number of available pixels varied along the season in high-latitude regions. To avoid misleading interpretation by the small number of samples with unreliable SSM reference, periods of less pixel availability (<30%) are grayed out.**




**Figure 5: Time series of global and zonal mean albedo. The left column shows the NIR band (A: global average; B–E: zonal average for each 30° in latitude), while the right column shows the visible band (the vertical arrangement is the same as that for NIR).**

**Figure 6: Boxplots of mean biases (simulated minus observed values) of SSM, ET, LAI (forced/coupled), and precipitation (coupled only) against each class of irrigation and precipitation. The upper limit, middle line, and lower limit of the boxes correspond 25-, 50- and 75- percentile values, respectively. The upper and lower limits of whiskers are maximum and minimum values, respectively. The diamond indicates mean value of the class. (A), (B): SSM bias; (C), (D): ET bias; (E), (F): LAI bias; (G), (H): coupled precipitation bias vs. irrigation and precipitation classes, respectively. Blue and pink boxes correspond to forced and coupled mode, respectively. The dashed lines indicate pixel availability (i.e., ratio of sampled pixels to all global land pixels) for each class. The horizontal black line shows zero. Each class of landscape factors is defined in Table 1.**





Figure 7: Boxplots of mean biases (simulated minus observed values) of ET, LAI, NIR/visible albedo (forced/coupled), and precipitation (coupled only) against each class of PFT and LAI. (A), (B): ET bias; (C), (D): LAI bias; (E), (F): NIR-albedo bias; (G), (H): visible-albedo bias; (I), (J): coupled precipitation bias vs. PFT and LAI classes, respectively. Legends and axes are the same as in Figure 6, and each class of landscape factors is defined in Table 1.








**Figure 8: Boxplots of mean biases (simulated minus observed values) against each class of ET, slope, and SWE. (A), (B): ET and LAI bias vs. ET class; (C), (D), (E): NIR albedo, visible albedo, and coupled precipitation bias vs. slope class; (F), (G): NIR- and visible-albedo bias against SWE class, respectively. Legends and axes are the same as in Figure 6, and each class of landscape factors is defined in Table 1.**
