# Peer review of "Multi-variable evaluation of land surface processes in forced and coupled modes reveals new error sources to the simulated water cycle in the IPSL climate model"

_Hydrology and Earth System Sciences, 2020_

## Referee Comment (RC1) · Eunkyo Seo (Referee) · 26 Nov 2020

This is a well written paper reporting the multi-variable evaluation of land surface processes in ORCHIDEE land surface model (land component of the IPSL climate model) under different simulation modes (either forcing by WFDEI or coupled with LMDZ). The research mainly covered that in-depth evaluation of four interlinked essential climate variables (e.g., surface soil moisture, evapotranspiration, leaf area index, and albedo) using various reference datasets (ESA CCI, upscaled FLUXNET, GIMSS 3g and MODIS products), and factor analysis is also conducted by several factors (e.g.,

plant functional type, leaf area index, irrigation, precipitation, slope, snow water equivalent, and evapotranspiration). I have several comments or suggestions for improvements and recommend publishing the paper after improving minor parts.

Minor comments

1) Line 120: there is specific information about the layer depth in the LSM. It is necessary to describe that information to notify which layer is defined as surface soil moisture.

2) Line 182: the comparison with both the forced and coupled simulations are conducted for 7 years (1993-1999: period 1). It would be clearer to compare the surface soil moisture against ESACCI and SMOS without dividing the period. If the forced simulation would be extended to 2014, that would be possible. The extending offline simulation seems to be possible because the essential datasets (e.g., ERA-Interim and monthly GPCC) used to force the land surface model are also available until 2014. Then, it seems that surface soil moisture is simply validated during 2010-2014.

3) Figure S5: it seems that the definition of subperiod 1 should be corrected to "1993-1999".

4) In many captions in main manuscript and supplementary, there is notation of "Temporally averaged". Does it mean "annual averaged"? It is necessary to be clear the validation season for each variable.

5) There is no description about the statistically significant test. It is needed in Section 2.

6) Figure 3: this figure represents the temporal correlation coefficient with monthly time series. Based on the explanation of Figure 4, the temporal correlation of surface soil moisture over 60N-90N is calculated by only using JJA monthly dataset. If it is correct, such a description should be added in the text.

7) In several Supplementary Figures, the information of the evaluation period and season for each dataset and variable is missed. It should be clearly notated.

Interactive
comment

8) Line 199: there is inconsistence in the evaluation period between the manuscript (1987-2009) and Figure S2 (1987-2006).

9) Lines 238-239: the impact of the chosen period would be marginal in terms of global mean aspect, but in the regional scale it is not likely to be limited.

10) Line 279: notation of "Figure 2E, F" is not matched to the description. It would be corrected to "Figure 2D, E".

11) Lines 290-291: the sentence of "the coupled ET was negatively biased" is not matched to Figure 2E in which the ET bias seems to be positive. In other words, the maritime region shows the positive feedback between precipitation and ET such as the Congo.

12) The positive bias of LAI is supported by the underestimation of LAI in the reference, but there is no discussion for the negative bias of the LAI. If the description is added, it would be helpful to understand the other aspect of the result.

13) If there is nothing of the description of the yearly variation, Figure 5 could be changed to showing the seasonal cycle only (e.g., Figure 4).

14) Figure 8B, Figure 6 E, F: the result of LAI mean bias in the forced and the coupled simulations is totally same to each other against each class even though there is difference in the non-classified result (notated to "all").

15) In the comparison of surface soil moisture with ESACCI and SMOS reference datasets, these two datasets have spatially and temporally different missing point. For even validation, even if any dataset is missed among them, all data should be excluded. Does it do that?

16) Line 414: the listed references for the SMAP product is not appropriate. Entekhabi et al. (2010) is the originated reference of the SMAP.

Entekhabi, D., Njoku, E. G., O'Neill, P. E., Kellogg, K. H., Crow, W. T., Edelstein, W. N.,
... & Kimball, J. (2010). The soil moisture active passive (SMAP) mission. Proceedings of the IEEE, 98(5), 704-716.

---

## Author Comment (AC1) · 8 Dec 2020

(Our response follows each original comment from the reviewer.)

This is a well written paper reporting the multi-variable evaluation of land surface processes in ORCHIDEE land surface model (land component of the IPSL climate model) under different simulation modes (either forcing by WFDEI or coupled with LMDZ). The research mainly covered that in-depth evaluation of four interlinked essential climate variables (e.g., surface soil moisture, evapotranspiration, leaf area index,

and albedo) using various reference datasets (ESA CCI, upscaled FLUXNET, GIMSS 3g and MODIS products), and factor analysis is also conducted by several factors (e.g.,plant functional type, leaf area index, irrigation, precipitation, slope, snow water equivalent, and evapotranspiration). I have several comments or suggestions for improvements and recommend publishing the paper after improving minor parts.

>Thank you very much for reviewing our manuscript. We are glad to receive your positive and constructive comments. Here we answer your comments one by one.

1) Line 120: there is specific information about the layer depth in the LSM. It is necessary to describe that information to notify which layer is defined as surface soil moisture.

>We used moisture in top 10 cm for ORCHIDEE SSM (Line 417). The information will be added in the revised manuscript.

2) Line 182: the comparison with both the forced and coupled simulations are conducted for 7 years (1993-1999: period 1). It would be clearer to compare the surface soil moisture against ESACCI and SMOS without dividing the period. If the forced simulation would be extended to 2014, that would be possible. The extending offline simulation seems to be possible because the essential datasets (e.g., ERA-Interim and monthly GPCC) used to force the land surface model are also available until 2014. Then, it seems that surface soil moisture is simply validated during 2010-2014.

>It is indeed conceivable to extend the offline simulation up to SMOS observation period using the latest WFDEI and GPCC datasets. However, extending the simulation also implies redoing all the evaluation work. It seems too laborious just to simplify the comparison scheme.

3) Figure S5: it seems that the definition of subperiod 1 should be corrected to "1993-1999".

>We will correct the definition in the revision, thank you.

4) In many captions in main manuscript and supplementary, there is notation of "Temporally averaged". Does it mean "annual averaged"? It is necessary to be clear the validation season for each variable.

>The "temporal average" was, if there is no annotation, taken during the entire evaluation period defined in Table 1. To clarify this, we will use "pluri-annual average" instead, in the revision.

5) There is no description about the statistically significant test. It is needed in Section 2.

>Both correlation coefficient and bias were tested by Student's t-test (also known test of no correlation and paired t-test, respectively) for each pixel. We will add the description in the revision.

6) Figure 3: this figure represents the temporal correlation coefficient with monthly time series. Based on the explanation of Figure 4, the temporal correlation of surface soil moisture over 60N-90N is calculated by only using JJA monthly dataset. If it is correct, such a description should be added in the text.

>Correlation coefficient was calculated using all available monthly time series for all seasons (not only JJA). The gray area in Figure 4 just shows less-reliable period and does not indicate data screening.

7) In several Supplementary Figures, the information of the evaluation period and season for each dataset and variable is missed. It should be clearly notated.

>We will add the description in the revision.

8) Line 199: there is inconsistence in the evaluation period between the manuscript (1987-2009) and Figure S2 (1987-2006).

>The description for Figure S1 (1987-2006) is incorrect, we will correct it to 1987-2009.

9) Lines 238-239: the impact of the chosen period would be marginal in terms of global mean aspect, but in the regional scale it is not likely to be limited.

>We will add this as a potential uncertainty or limitation of our analysis in Discussion.

10) Line 279: notation of "Figure 2E, F" is not matched to the description. It would be corrected to "Figure 2D, E".

>We will correct it, thank you.

11) Lines 290-291: the sentence of "the coupled ET was negatively biased" is not matched to Figure 2E in which the ET bias seems to be positive. In other words, the maritime region shows the positive feedback between precipitation and ET such as the Congo.

>There is a hot-spot where ET is negatively biased in the maritime region (provided Fig. 1), although it is a bit difficult to see it in this resolution of the figure (the originally provided one has more fine resolution). But as you pointed out, the other part of the maritime region shows positive bias, and we will make the description more precise in the revision.

12) The positive bias of LAI is supported by the underestimation of LAI in the reference, but there is no discussion for the negative bias of the LAI. If the description is added, it would be helpful to understand the other aspect of the result.

>This comment refers to Lines 296-300, where it is true that the negative LAI biases are not discussed. Eastern Siberia is the main place with negative LAI biases, and we will mention possible explanations, which are consistent with Guimberteau et al. (2018): too persistent snowpack reducing the growth season, consistent with positive albedo biases in some parts of eastern Siberia; underestimated maximum LAI in the model; errors in the reference LAI product, especially at high latitudes, like for albedo. We will also send to the Discussion section, where some negative bias areas are already addressed, at Lines 353 (irrigated areas), 356 (interaction with negative ET and SSM bias), 394 (Himalayas in coupled mode), and 400 (Eastern Siberia).

13) If there is nothing of the description of the yearly variation, Figure 5 could be

changed to showing the seasonal cycle only (e.g., Figure 4).

>We will change the Figure 5 to show the seasonal cycle only.

14) Figure 8B, Figure 6 E, F: the result of LAI mean bias in the forced and the coupled simulations is totally same to each other against each class even though there is difference in the non-classified result (notated to "all").

>We apologize that there were some errors in making the final figures (forced plots came to be the same as coupled ones accidentally). Actually, coupled plot tended to be lower than forced one (provided Fig.2), which is consistent with the difference in the "all" category. It does not change the conclusion because we had made discussion based on the right figures. We will correct the figure in the revised version.

15) In the comparison of surface soil moisture with ESACCI and SMOS reference datasets, these two datasets have spatially and temporally different missing point. For even validation, even if any dataset is missed among them, all data should be excluded. Does it do that?

>When different missing points were included in the different product during the same period, both data at the point were excluded in the analysis. We referred to it as "co-masking" in the manuscript (Line 231).

16) Line 414: the listed references for the SMAP product is not appropriate. Entekhabi et al. (2010) is the originated reference of the SMAP. Entekhabi, D., Njoku, E. G., O'Neill, P. E., Kellogg, K. H., Crow, W. T., Edelstein, W. N., C3 HESSD Interactive comment Printer-friendly version Discussion paper ... & Kimball, J. (2010). The soil moisture active passive (SMAP) mission. Proceedings of the IEEE, 98(5), 704-716.

>Thank you for the information. We will add your suggested reference, but also keep the two references as they are useful when using multiple SSM products.

[Figure]

438, 2020.

**Fig. 1.** Zoomed ET bias in the original Figure 2-E

[Figure]

**Fig. 2.** Corrected LAI figures

---

## Referee Comment (RC2) · Anonymous Referee #2 · 19 Dec 2020

Mizuochi and co-authors present a multivariate evaluation of the IPSL climate model forced with and without coupling to GCM at the global scale. Number of independent products are used to evaluate the performance of four Essential Climate Variables plus precipitation between the coupled and uncoupled modes and assess their differences. Although I am not an expert in the ORCHIDEE/LSM modelling, the manuscript is mostly clearly written, storyline is nicely motivated. Some rearrangements are recommended (see further), to better quantity different model configurations (factor assessment) and their linkage. Overall, the presented topic is interesting and relevant to be the HESS

readership. Note that the model simulations are not available for reviewer assessment, which does not comply with Copernicus guidelines, see https://www.hydrology-and-earth-system-sciences.net/policies/data$_{policy.html for more details}$.

**Main comments:**

(1) I have missed some extensive quantification and discussion, how much precipitation differs between WFDEI and LMDZ6A atmospheric GCM (period, annual, seasonal values) at the beginning of your analysis. There is panel C in Figure 2 introduces "coupled precipitation bias", but it is not defined, what is the references for this bias. Is it WFDEI or GPCC? It might be also helpful to introduce, how much this error is in terms of the relative annual change, to get better feeling for their differences. Which one of the two is closer to reality?

(2) It is quite unusual that you introduce first three figures in the first paragraph of the Results section. Please, better explain the sequence of Figures and clearly explain, why you are showing all the figures. It is quite needed to make a better link between fig. 1 (depicting the factors) and following figures, especially the classes of Figures 6-8 need to be better linked. Current x-axes definitions of Fig. 6-8 is missing and need to be clearly linked to Table 2.

(3) There is positive bias for the coupled ET (see Figure 2E) in Amazonas, while the coupled precipitation bias (Figure 2C) shows large precipitation underestimation for the very same region. I don't understand, where this counter-intuitive behavior comes from. Please clarify.

(4) Figure 2 is already quite heavy, but it is really hard to spot the difference between left and right column. Would be nice to, say, provide the relative change between the forced and coupled runs.

**Minor/Technical:**

Remove reference in preparation

Line 264: spell out CC is correlation coefficient (Pearson I guess?)

Fig. 1D, not clear, why authors use do you use the ORCHIDEE variable in this reference overview figure and not an independent data source.

Fig. 1G, link types to Table 1.

Figure 3: I suggest, replace gray by white and then green by gray. Green takes too much attention, although that's non-significant pixels.

---

## Author Comment (AC2) · 26 Jan 2021

Mizuochi and co-authors present a multivariate evaluation of the IPSL climate model forced with and without coupling to GCM at the global scale. Number of independent products are used to evaluate the performance of four Essential Climate Variables plus precipitation between the coupled and uncoupled modes and assess their differences. Although I am not an expert in the ORCHIDEE/LSM modelling, the manuscript is mostly clearly written, storyline is nicely motivated. Some rearrangements are recommended (see further), to better quantity different model configurations (factor assessment) and

their linkage. Overall, the presented topic is interesting and relevant to be the HESS readership. Note that the model simulations are not available for reviewer assessment, which does not comply with Copernicus guidelines, see https://www.hydrology-andearth-system-sciences.net/policies/datapolicy.htmlformoredetails.

>Thank you very much for reviewing our paper. We are glad to read your encouraging comments. Hereby we provide one-to-one reply as follows, and the manuscript will be revised based on your suggestions in the revision stage. As for the original monthly simulation data, it will be made accessible through the IPSL web platform, by both anonymous ftp and http/opendap protocols at:

> ftp.climserv.ipsl.polytechnique.fr/Data4papers/

> https://vesg.ipsl.upmc.fr/thredds/catalog/IPSLFS/datapapers

>IPSL will also issue a doi and the corresponding metadata will be documented through a catalogue (https://data.ipsl.fr/catalog/).

Main comments: (1) I have missed some extensive quantification and discussion, how much precipitation differs between WFDEI and LMDZ6A atmospheric GCM (period, annual, seasonal values) at the beginning of your analysis. There is panel C in Figure 2 introduces "coupled precipitation bias", but it is not defined, what is the references for this bias. Is it WFDEI or GPCC? It might be also helpful to introduce, how much this error is in terms of the relative annual change, to get better feeling for their differences. Which one of the two is closer to reality?

>"The coupled precipitation bias" is the pluri-annual mean difference (i.e. simulation minus reference) from 1987 to 2009 (Table 1) in the coupled simulation, using GPCC as reference. To clarify this, we propose to recognize precipitation as a fifth evaluated variable, and to provide the corresponding bias for both simulations forced and coupled (Figure 2, see new version below, and Tables 3, 4).

>It is noteworthy that the new Figure 2A shows a precipitation bias for the forced simulation which is not everywhere negligible (+0.112 mm/d on average over land, compared to +0.186 mm:d for the coupled simulation). This contradicts our initial assumption, motivated by the fact that GPCC was also used for bias-correction of the WFDEI precipitation used to force ORCHIDEE (Line 153). The forced ORCHIDEE precipitation (WFDEI) is higher than GPCC in small tropical pockets, the US Great Plains, and boreal zones, which are prone to precipitation undercatch because of strong winds and/or a large fraction of snowfall (Becker et al., 2013). Schneider et al. (2014) acknowledge, for the GPCC product, that "the biggest uncertainty issue is the correction of the systematic gauge-measuring error (general undercatch of the true precipitation)", but this is very likely true for all precipitation products.

>The difference between the forced ORCHIDEE precipitation (WFDEI) and GPCC thus probably comes from their different methods to correct undercatch: based on Legates and Willmott (1990) for GPCC, and on Adam and Lettenmeier (2003) for WFDEI. Based on the literature, we are not able to assess which product is the most realistic, but we believe that a clear discussion of this issue will be useful to the HESS readers, in general and to better put in perspective the biases of the other variables. The above elements will therefore be inserted in the revised manuscript (in sections 2.1.2, 2.2.5, and 3.1) and lead also to slight changes in sections 3.2 and 4). For the last part of your comment, we did not quite understand the suggestion, but we hope the above-mentioned clarifications are enough.

>To finish with precipitation, we propose to specify in the revised manuscript that a few changes have been made in physics of the atmosphere between the version of the IPSL climate model used in this paper and the one used for CMIP6: "The only difference between the atmospheric physics used in this paper and the one used for CMIP6 concern the parameterization of deep and shallow convection and their interaction to improve the description of the intertropical convergence zone and the El Nino Southern Oscillation. The description of these differences and their impact on precipitation and other variables controlling the near surface climate can be found in Mignot et al.

(2021)."

(2) It is quite unusual that you introduce first three figures in the first paragraph of the Results section. Please, better explain the sequence of Figures and clearly explain, why you are showing all the figures. It is quite needed to make a better link between fig. 1 (depicting the factors) and following figures, especially the classes of Figures 6-8 need to be better linked. Current x-axes definitions of Fig. 6-8 is missing and need to be clearly linked to Table 2.

>We first introduce the pluri-annual mean maps for all the variables for reference products, forced and coupled simulations (in Supplementary), but the first paragraph announces that we rather focus on the bias maps which are more synthetic (Fig. 2). This paragraph also announces a summary table and a few maps of the correlation coefficient between simulated variables and reference products because the rest of Section 3.1 organizes the analysis by variable (precipitation, SSM, ET, LAI, and albedo) and not by type of performance criteria (bias and coefficient of correlation). We will also update the caption of Fig.1 to better explain all reference data (see the revised Figure at the end of this response). Some of them were used for validation (indicated as "+" symbols in the caption), the others were used for factor analysis (indicated as "x" symbols). Figs. 6-8 are the results of the factor analysis, along the x-axis classes defined in Table 2. This point is mentioned at the end of the captions. For example, Fig.6a (irrigation panel for SSM bias) represents distribution of SSM bias patterns for each irrigation class, defined in Table 2: from small (0%; class 1) to large (50-100%; class 6) fractions of irrigated area.

(3) There is positive bias for the coupled ET (see Figure 2E) in Amazonas, while the coupled precipitation bias (Figure 2C) shows large precipitation underestimation for the very same region. I don't understand, where this counter-intuitive behavior comes from. Please clarify.

>We have already commented this counter-intuitive result in the submitted manuscript

(Lines 288-292 - Results, 369-370 - Discussion). The suggested explanation involves a negative feedback in the region between precipitation and ET because the area is strongly energy-limited (Seneviratne et al., 2010; McVicar et al., 2012). As a result, excessive precipitation does not induce excessive ET as ET is not water limited, but the related excessive cloudiness drives a deficit of incoming solar radiation, which likely explains the negative ET bias.

(4) Figure 2 is already quite heavy, but it is really hard to spot the difference between left and right column. Would be nice to, say, provide the relative change between the forced and coupled runs.

>We provide below the figures showing difference between forced and coupled modes (Revised supplementary fig. S6). However, the original figures are also useful to indicate errors in each mode, our paper is organized to use them (see above, point 3) and we would like to keep them too. We will thus provide the difference panels in the supplementary, and announce it in the introductory paragraph of section 3.1.

Minor/Technical: Remove reference in preparation

>We will remove it from the revised manuscript.

Line 264: spell out CC is correlation coefficient (Pearson I guess?)

>As you guessed, it is Pearson's correlation coefficient. We will mention it in the revised manuscript.

Fig. 1D, not clear, why authors use do you use the ORCHIDEE variable in this reference overview figure and not an independent data source.

>Fig 1D shows the mean snow water equivalent simulated by ORCHIDEE offline. It is not used as a reference for validation, but for the factor analysis. In other words, it was simply used to separate less snowy area from more snowy area in the additional analysis after the fundamental validation. Independent data source is essentially needed for the validation, but we assumed it is not necessarily required in such a classification

purpose. We will add the explanation in the revised manuscript. We will also clarify the caption of Fig. 1, using symbols to distinguish the variables that are used for validation (+) and/or factor analysis (x).

Fig. 1G, link types to Table 1.

>The reviewer probably meant to link with the PFT classes of Table 2, and it is a very good suggestion. We will indeed add in the caption of Fig. 1 that the PFT classes are defined in Table 2.

Figure 3: I suggest, replace gray by white and then green by gray. Green takes too much attention, although that's non-significant pixels.

>Thank you for this useful suggestion. We changed the colors in the revised Fig. 4 below, and we will use thus new figure in the revised manuscript.

>Cited references:

>Adam, J. C., and Lettenmaier, D. P.: Adjustment of global gridded precipitation for systematic bias. J. Geophys. Res., 108, 4257, doi:10.1029/2002JD002499, 2003.

>Becker, A., Finger, P., Meyer-Christoffer, A., Rudolf, B., Schamm, K., Schneider, U., and Ziese, M.: A description of the global land-surface precipitation data products of the Global Precipitation Climatology Centre with sample applications including centennial (trend) analysis from 1901–present, Earth Syst. Sci. Data, 5, 71–99, https://doi.org/10.5194/essd-5-71-2013, 2013.

>Legates, D.R., and Willmott, C.J.: Mean seasonal and spatial variability in gauge-corrected, global precipitation. Int. J. Climatol. 10, 111–127, http://dx.doi.org/10.1002/joc.3370100202, 1990.

>Mignot, J., Hourdin, F., Deshayes, J., Boucher, O., Gastineau, G., Musat, I., Vancoppenolle, M., Servonat, J., Caubel, A., Cheruy, F., Denvil, S., Dufresne, J.-L., Ethe, C., Fairhead, L., Foujols, M.-A., Grandpeix, J.-Y., Levavasseur, G., Marti, O., Menary, M.,

Rio, C., Rousset, C.: The tuning strategy of IPSL-CM6A-LR, JAMES, revision submitted.

>Schneider, U., Becker, A., Finger, P., Meyer-Christoffer, A., Ziese, M., and Rudolf, B.: GPCC's new land surface precipitation climatology based on quality-controlled in situ data and its role in quantifying the global water cycle. Theoretical and Applied Climatology, 115(1–2), 15–40, https://doi.org/10.1007/s00704-013-0860-x, 2014.

———————————

[Figure]

**Fig. 1.** Revised Figure 2: Temporally-averaged spatial patterns of bias (i.e., simulated values minus observed values) for the five evaluated variables, simulated in forced mode (left) and coupled mode (right)

[Figure]

**Fig. 2.** Revised supplementary figure S6. Pluri-annual mean relative differences between forced and coupled modes for normalized SSM, ET, LAI, and albedo, in % of the forced value

[Figure]

**Fig. 3.** Revised Figure 1: Spatial patterns of pluri-annual mean reference data used for validation (marked with a "+") and/or factor analysis (marked with a "x")

[Figure]

**Fig. 4.** Revised Figure 4. Color-changed version of the original Fig. 3.